# Differential Gene Expression Associated with Soybean Oil Level in the Diet of Pigs

**DOI:** 10.3390/ani12131632

**Published:** 2022-06-25

**Authors:** Simara Larissa Fanalli, Bruna Pereira Martins da Silva, Julia Dezen Gomes, Vivian Vezzoni de Almeida, Felipe André Oliveira Freitas, Gabriel Costa Monteiro Moreira, Bárbara Silva-Vignato, Juliana Afonso, James Reecy, James Koltes, Dawn Koltes, Luciana Correia de Almeida Regitano, Dorian John Garrick, Júlio Cesar de Carvalho Balieiro, Ariana Nascimento Meira, Luciana Freitas, Luiz Lehmann Coutinho, Heidge Fukumasu, Gerson Barreto Mourão, Severino Matias de Alencar, Albino Luchiari Filho, Aline Silva Mello Cesar

**Affiliations:** 1Faculty of Animal Science and Food Engineering, University of São Paulo, Pirassununga 13635-900, SP, Brazil; simarafanalli@usp.br (S.L.F.); brunamartins@usp.br (B.P.M.d.S.); fukumasu@usp.br (H.F.); 2Luiz de Queiroz College of Agriculture, University of São Paulo, Piracicaba 13418-900, SP, Brazil; juliadezen@usp.br (J.D.G.); felipeoliveirafreitas@usp.br (F.A.O.F.); barbarasilva@usp.br (B.S.-V.); arimeira@gmail.com (A.N.M.); llcoutinho@usp.br (L.L.C.); gbmourao@usp.br (G.B.M.); smalencar@usp.br (S.M.d.A.); luchiari@usp.br (A.L.F.); 3College of Veterinary Medicine and Animal Science, Federal University of Goiás, Goiânia 74690-900, GO, Brazil; vivian.almeida@ufg.br; 4GIGA Medical Genomics, Unit of Animal Genomics, University of Liège, 4000 Liège, Belgium; gcmmoreira@gmail.com; 5Embrapa Pecuária Sudeste, São Carlos 70770-901, SP, Brazil; juafonsobio@gmail.com (J.A.); luciana.regitano@embrapa.br (L.C.d.A.R.); 6College of Agriculture and Life Sciences, Iowa State University, Ames, IA 50011, USA; jreecy@iastate.edu (J.R.); jekoltes@iastate.edu (J.K.); delkins@iastate.edu (D.K.); 7AL Rae Centre for Genetics and Breeding, Massey University, Hamilton 3214, New Zealand; d.garrick@massey.ac.nz; 8College of Veterinary Medicine and Animal Science, University of São Paulo, Pirassununga 13635-900, SP, Brazil; juliobalieiro@gmail.com; 9DB Genética de Suínos, Patos de Minas 38706-000, MG, Brazil; luciana@db.agr.br

**Keywords:** fatty acid, RNAseq, transcriptome, immune response, metabolism, hepatic tissue, *Longissimus lomborum*, animal model

## Abstract

**Simple Summary:**

Findings from the analysis of the pig transcriptome may help to better understand the biological mechanisms that can be modulated by the diet. Thus, the aim of this study was to identify the differentially expressed genes from the skeletal muscle and liver samples of pigs fed diets with two different levels of soybean oil (1.5 or 3%). The FA profile in the tissues was modified by the diet mainly related to monounsaturated (MUFA) and polyunsaturated (PUFA). This nutrigenomics study verified the effect of different levels of soybean oil in the pig diet on the transcriptome profile of skeletal muscle and liver, where the higher level of soybean oil added to the diet led to a higher expression of genes targeting biological processes related to lipid oxidation and consequently to metabolic diseases and inflammation.

**Abstract:**

The aim of this study was to identify the differentially expressed genes (DEG) from the skeletal muscle and liver samples of animal models for metabolic diseases in humans. To perform the study, the fatty acid (FA) profile and RNA sequencing (RNA-Seq) data of 35 samples of liver tissue (SOY1.5, *n* = 17 and SOY3.0, *n* = 18) and 36 samples of skeletal muscle (SOY1.5, *n* = 18 and SOY3.0, *n* = 18) of Large White pigs were analyzed. The FA profile of the tissues was modified by the diet, mainly those related to monounsaturated (MUFA) and polyunsaturated (PUFA) FA. The skeletal muscle transcriptome analysis revealed 45 DEG (FDR 10%), and the functional enrichment analysis identified network maps related to inflammation, immune processes, and pathways associated with oxidative stress, type 2 diabetes, and metabolic dysfunction. For the liver tissue, the transcriptome profile analysis revealed 281 DEG, which participate in network maps related to neurodegenerative diseases. With this nutrigenomics study, we verified that different levels of soybean oil in the pig diet, an animal model for metabolic diseases in humans, affected the transcriptome profile of skeletal muscle and liver tissue. These findings may help to better understand the biological mechanisms that can be modulated by the diet.

## 1. Introduction

The World Health Organization [1] estimated that 41 million people died in 2018 due to chronic non-communicable diseases (NCDs) including cardiovascular diseases (CVDs), cancers, and metabolic diseases such as type 2 diabetes, obesity, and neurodegenerative diseases. These diseases have in common the influence of genetic factors, oxidative stress, and inflammation as result of sedentary lifestyle, diet, and consumption of drugs and alcohol. Pigs have been used as an animal model for diseases and in nutrigenetics and nutrigenomics [2,3,4,5].

Metabolic diseases are any of the diseases or disorders that disrupt the normal process of converting food to energy into the cell, which involves the activity of thousands of enzymes. Among the metabolic processes that occur in the cell are the process and transport of the proteins, carbohydrates, lipids, and their small organic molecules such as amino acids, sugars and starches, and fatty acids (FA), respectively [6,7].

The main process that generates energy is cellular respiration that uses organic molecules (sugar and lipids) and oxygen. Part of the oxygen used in this process is transformed into superoxide or reactive oxygen species (ROS) that can cause harm to cells, tissues, and organs [7]. Free radicals such as ROS are highly toxic and reactive molecules capable of transforming other molecules such as proteins, lipids, and nucleic acids (DNA and RNA). This transformation changes the chemical structure and biological function of these molecules, disrupting the normal cellular metabolic processes and, as a consequence, leads to an inflammatory state and a pro-inflammatory immune response [6].

Meat has high nutritional importance and FA profile has impact on human health. Among its many benefits are its supply of unsaturated fatty acids, such as oleic acid (AO, C18:1 cis 9) and linoleic acid (AL, C18:2 cis9, 12), which have positive effects on human health [8]. Pigs have been increasingly used in nutrigenomics research and for the investigation of metabolic diseases in humans due to their role as an animal model for humans [2,5,9]. Pigs have similarities in terms of anatomy, biochemistry, pathology, pharmacology, and physiology with humans which causes them to be used as an animal model [3,9,10,11].

In pigs, the liver is a highly specialized organ, related to the regulation of several metabolic processes, along with the skeletal muscle, which is essential for the regulation of lipid metabolism in mammals [12]. Muscle tissue is of fundamental importance in regulating lipid metabolism, which is regulated by various factors and molecules [13]. Free fatty acids are released from adipose tissue and enter skeletal muscle via intermembrane proteins. In addition, free fatty acids can regulate several genes and skeletal muscle lipogenesis, including sterol regulatory element-binding proteins (*SREPB*), nuclear factor kappa B (*NFkB*), liver X receptors, retinoid X receptors, and receptors activated by peroxisome proliferators (*PPARs*) [13].

The characterization of gene networks involved in immune cell function and metabolism in health and disease is important to understand immunodeficiency or autoimmunity disorders caused by an unbalanced immune response [14]. Dysfunctional peroxisomes result in overall lipid alterations, such as the accumulation of very-long-chain fatty acids (VLCFA) and VLCFA-cholesteryl esters [14]. Lipid metabolic dysregulation can cause inflammation, participating in the modulation of the immune system responses [14,15,16]. Furthermore, among the proteins that drive inflammation are cytokines that are previously secreted by immune cells and vascular endothelial cells [14,15].

Lipids are a major class of biological molecules, ubiquitously distributed in all types of cells. They regulate transcription, store energy, and contribute to many biological processes, such as cellular structure and energy storage [17]. FA are among the most important dietary components found in meat, which is rich in polyunsaturated (PUFA), saturated (SFA), and monounsaturated (MUFA) fatty acids [18]. The consumption of FA has been linked to metabolic effects, including altered blood parameters such as lipid and lipoprotein composition [18]. Nevertheless, past studies and meta-analyses have noted contrasting opinions regarding the role of FA in animal and human health [19].

Our previous studies in different species revealed a significant number of genes differentially expressed between animals with divergent values of FA content in skeletal muscle, associated with biological processes such as insulin receptor signaling, activated T cell nuclear factors (NFAT) in cardiac hypertrophy, mitochondrial disorder, and neurodegenerative disorders such as Huntington’s and Alzheimer’s disease (AD). The study shows that fatty acids have a significant impact on gene expression associated with important biological processes, such as oxidative phosphorylation, cell growth, survival, and migration [20].

In relation to MUFA or PUFA consumption, metabolic studies presented in Schmid [21] point out that these FA are associated with positive effects on the total cholesterol and high-density lipoprotein (HDL) ratio, in which these FA promote a reduction in low-density lipoprotein (LDL) and an increase in HDL in the blood. Pig meat has a high unsaturated fatty acid profile, mainly due to oleic acid (OA), and plays a significant role in human nutrition and health [21,22,23,24]. Another important aspect of unsaturated FA content is the relationship with meat quality characteristics, such as juiciness, flavor, and shelf life [25]. In the context of improving our knowledge about the biological processes associated with FA content in different tissues, scientific studies of this kind are extremely important. This knowledge can be applied both in animal production (genetic improvement, nutrition, and environment) and in animal and human health.

Pig diets with added vegetable oils rich in unsaturated fatty acids can be a source of healthier products for consumers [26]. Thus, vegetable oils such as canola, sunflower, and soybean oil are interesting options due to the amount of PUFA [27]. On the other hand, there is a need to fill in the knowledge gaps regarding the impact of the amount of soybean oil added on gene expression and the functional and nutritional knowledge of lipids. Thus, our hypothesis is that different levels of soybean oil (1.5% vs. 3%) added in the diet of Large White pigs can affect the variation in the fatty acid profile deposition, which could be associated with the gene expression profile of the skeletal muscle and liver tissue of these animals, modulating biological processes involved with lipid metabolism and metabolic diseases.

The present study aimed to (1) evaluate changes in fatty acid profile and gene expression of the skeletal muscle and liver tissue of pigs fed a diet with different levels of soybean oil; (2) identify metabolic pathways and gene networks impacted by the dietary fatty acid composition of the pigs’ tissues that were fed with different proportions of degummed soybean oil.

## 2. Materials and Methods

### 2.1. Ethics Statement

All procedures involving animals were approved by the Animal Care and Use Committee of Luiz de Queiroz College of Agriculture (University of São Paulo, Piracicaba, Brazil, protocol: 2018.5.1787.11.6 and number CEUA 2018-28) and followed ethical principles in animal research, according to the Guide for the Care and Use of Agricultural Animals in Agricultural Research and Teaching [28].

### 2.2. Animals and Diets

In our 98-day feeding study, thirty-six purebred immunocastrated male pigs (offspring of Large White sires × Large White dams) with an average initial body weight (BW) of 28.44 ± 2.95 kg and an average age of 71 ± 1.8 days were used. All the animals were genotyped for the halothane mutation (*RYR1* gene) by molecular test according to Fujii et al. [29]; thus, pigs selected for this trial were all halothane-free (NN). The animals were randomly allotted to one of two dietary treatments with six replicate pens per treatment and three pigs per pen, which were housed in an all-in/all-out double-curtain-sided building. Each pen was equipped with a three-hole dry self-feeder and a nipple drinker, allowing pigs ad libitum access to feed and water throughout the experimental period. Immunocastration of the intact males was performed by the administration of two 2 mL doses of Vivax^®^ (Pfizer Animal Health, Parkville, Australia) on fatting day 56 (127 days of age) and fatting day 70 (141 days of age) [30,31], in accordance with the manufacturer’s recommendations. The study was performed at experimental farm of DB Genética company, and the animals were slaughtered at 155 days of age on average.

The experimental diet consisted of a six-phase diet that was as follows: two in the grower and four for finisher [30]. Dietary treatments consisted of corn–soybean meal growing–finishing diets supplemented with 1.5% soybean oil (SOY1.5, common level used in commercial pig production) or 3% soybean oil (SOY3.0). The diets were formulated to reach or exceed Rostagno et al. [32] recommendations for growing–finishing pigs. The diets were formulated to have a similar level of digestible energy. No antibiotic growth promoters were used, and all diets were provided in a mash meal form; details of animals and diets of this study are described in Appendix A, which were adapted from our previous study [30].

### 2.3. Fatty Acid Profile

Sample collection fatty acid profile was previously described in Almeida et al. [30] and Fanalli et al. [31]. In summary, liver and skeletal muscle (*Longissimus lumborum*) samples were collected, and then stored at −80 °C until fatty acid profile and RNA sequencing analyses. The FA profile was performed by Bligh and Dyer [33] and methylated according to the procedure outlined by AOCS [34] (Method AM 5-04).

Statistical analyses to verify differences in the FA profile of skeletal muscle and liver tissue between the diets were performed using the “proc mixed” procedure of the SAS statistical software (v. 9.4), where the mixed model was adopted using the restricted maximum likelihood (REML) methodology. In the model, the block effects were declared as a random effect and the treatments as a fixed effect. A normal distribution of the data was assumed and exploratory analyses were previously performed to verify the consistency of the data. The SAS “proc univariate” procedure (v. 9.4) was used to verify the fit of the normal distribution and homogeneity of residuals for each of the variables. Diagnostics of the density distribution of the studentized residual of the model were made with the Shapiro–Wilk test and, also, graphs were plotted as a histogram with normal density, scatterplot and “QQ plot” for visual analysis of the dispersion of residues with the option “residual” of the “mixed proc” (SAS v. 9.4).

### 2.4. Tissue RNA Extraction and RNA Sequencing

Total RNA was extracted from skeletal muscle and liver tissue samples using commercial RNA extraction kits (RNeasy^®^ Mini Kit, Qiagen, Hilden, Germany), according to the manufacturer’s instructions. With the spectrophotometer Nanodrop 1000 and Bioanalyzer, RNA quantification, purity, and integrity were evaluated, respectively. All samples presented an RNA integrity number (RIN) higher or equal to seven (Appendix A). From the total RNA from each sample, 2 µg was used for library preparation according to the protocol described in the TruSeq RNA Sample Preparation kit v2 guide (Illumina, San Diego, CA, USA). The estimation of libraries’ average size was made with the Agilent Bioanalyzer 2100 (Agilent, Santa Clara, CA, USA) and the libraries were quantified using quantitative PCR with the KAPA Library Quantification kit (KAPA Biosystems, Foster City, CA, USA). Quantified samples were diluted and pooled (five pools of all 36 samples each), using TruSeq DNA CD Index Plate (96 indexes, 96 samples, Illumina, San Diego, CA, USA). All samples were pooled and sequenced in five lanes of a sequencing flow cell, using the TruSeq PE Cluster kit v4-cBot-HS kit (Illumina, San Diego, CA, USA), and were clustered and sequenced using HiSeq2500 equipment (Illumina, San Diego, CA, USA) with a TruSeq SBS Kit v4-HS (200 cycles), according to manufacturer instructions. All the sequencing analyses were performed at the Genomics Centre at ESALQ, localized in the Animal Biotechnology Laboratory at ESALQ—USP, Piracicaba, São Paulo, Brazil [20].

Sequencing adaptors and low-complexity reads were removed in an initial data-filtering step by Trim Galore 0.6.5 software.Only the reads with a length higher than 70 nucleotides and a Phred score lower than 33 were kept after trimming. Quality control and reads statistics were estimated with FASTQC version 0.11.8 software [http://www.bioinformatics.bbsrc.ac.uk/projects/fastqc/] (accessed on 15 July 2021). *Sus scrofa 11.1* reference assembly available at Ensembl [http://www.ensembl.org/Sus_scrofa/Info/Index] (accessed on 15 July 2021). The abundance (read counts) of mRNAs for all annotated genes was calculated using STAR-2.7.6a [http://bioinformatics.oxfordjournals.org/content/29/1/15] (accessed on 15 July 2021).

### 2.5. Data Analysis, Differentially Expressed Genes, and Functional Enrichment Analysis

Differentially expressed genes (DEG) between the two different diets (SOY1.5 and SOY3.0) from skeletal muscle and liver tissue were identified using the DESeq2 available at Bioconductor open-source software for bioinformatics, using a multi-factor design [35] statistical package in R. Prior to statistical analysis, the read count data were filtered as follows: (i) genes with zero counts for all samples, that is, unexpressed genes; (ii) genes with less than one read per sample on average were removed (very lowly expressed); (iii) genes that were not present in at least 50% of the samples were removed (rarely expressed). Sire was fit as factor in the multi-factor model. The cut-off approach performed to identify the DEG was control the false discovery rate (FDR) at 10% according to previous studies and DESeq2 recommendations [20,35], by using the Benjamini–Hochberg [36] methodology.

For skeletal muscle, the functional enrichment analysis by MetaCore [37] was applied to identify the pathway maps from 45 DEG, and in the liver 281 DEG. The functional enrichment analysis of DEG (FDR < 0.10) was performed to obtain comparative networks by ‘analyze single experiment’ using a standard parameter of MetaCore software v.22.1 build 70,800 using *Homo sapiens* genome annotation as background reference and a default parameter. The filters used were metabolic maps: energy metabolism, lipid metabolism, steroid metabolism; cardiovascular diseases: atherosclerosis; regulation of metabolism; nutritional and metabolic diseases, and nervous system diseases. To understand the behavior of genes and their interactions, gene networks were created using the Process Networks tool in MetaCore.

## 3. Results

### 3.1. Fatty Acid Profile for Skeletal Muscle and Liver Tissue

To analyze the changes in FA profiles, we identified the FA composition in the tissues, resulting in lipidic profile changes in skeletal muscle and liver of growing and finishing pigs receiving a diet enriched with different proportions of soybean oil.

The *Longissimus lumborum* intramuscular fat composition (Table 1) was modified with the increase in the proportion of oil to C18:1 n-9) (*p*-value < 0.01); that is, with the increase in dietary oil, FA deposition increased.

In addition, the composition of the FA profile was also performed in the liver. The FA profile in the liver (Table 2) presented a different pattern to that observed in the skeletal muscle in relation to the SFA as the C14:0 and C16:0 that presented differences (*p*-value < 0.01), with a high percentage in the samples from animals that received a diet enriched with 3% of oil, which can be due to the increase in palmitic and myristic acid. The C18:0 presented a higher content in samples from animals with the diet enriched in SOY1.5 (*p*-value < 0.01). As it occurred in the muscle, high values of OA were identified in the liver, C18:2 n-6 (*p*-value < 0.01) was higher in SOY1, and C18:3 n-3 did not present a difference between diets (*p*-value = 0.07). The C22:6 n-3 presented a difference in intramuscular fat (*p*-value = 0.03) and not in the liver (*p*-value = 0.11). The atherogenic index was lower in SOY1.5 compared to SOY3.0 (*p*-value < 0.01).

### 3.2. Sequencing Data and Differential Expression Analysis

Thus, 36 samples (SOY1.5 vs. SOY3.0) of skeletal muscle and 35 samples (17 of SOY1.5 and 18 of SOY3.0) of liver tissue were used in subsequent RNA-sequencing (RNA-Seq) data analysis. The total average number of sequenced reads before and after filtering for samples from the skeletal muscle of SOY1.5 group was 33,459,142 and 32,965,842, and of SOY3.0 group were 31,955,613 and 31,491,236. The total average number of reads before and after filtering for samples from the liver tissue of the SOY1.5 group was 33,561,721 and 33,072,908, and of the SOY3.0 group were 34,078,903 and 33,610,858. On average, 78.59% of the total read pairs were uniquely mapped against the *Sus scrofa 11.1* reference genome assembly (Appendix A) for both tissues and for each treatment.

Differential gene expression analysis was performed for each tissue by comparing gene expression levels between the two groups of animals that were fed with diets enriched with different levels of soybean oil (SOY1.5 vs. SOY3.0). A total of 45 DEG (log2 fold change ≥1; ≤−1; FDR-corrected *p*-value < 0.10) were identified in the skeletal muscle between the two groups, where 35 were down-regulated (log2-fold change ranging from −5.8 to −0.41) and 10 up-regulated (log2-fold change ranging from 2.3 to 0.53) in the SOY1.5 group compared with the SOY3.0 (Appendix A). For liver tissue, a total of 281 DEG (log2-fold change ≥1; ≤−1; FDR-corrected *p*-value < 0.10) were identified, where 129 were down-regulated (log2FC ranging from −3.0 to −0.20) and 152 up-regulated (log2FC ranging from 4.8 to 0.24) in the SOY1.5 group (Appendix A). Appendix A shows the volcano plot of the log2-fold change (x-axis) vs. the log10FDR-corrected *p*-value (y-axis) for skeletal muscle (Appendix A) and liver tissue (Appendix A) from the differential gene expression analysis between comparisons and a heatmap of the count matrix using transformed data for skeletal muscle (Appendix A) and liver tissue (Appendix A).

### 3.3. Common Differentially Expressed Genes between Skeletal Muscle and Liver Tissue

Among the DEG identified in the skeletal muscle and liver tissue (Table 3) of animals fed with the two different levels of soybean, six of them were common. One of them was the *CDK20* gene, which was higher expressed in the SOY1.5 group for both tissues (log2-fold change +1.04 and +0.98, respectively). In the same way, the *CCDC90B* was less expressed in the SOY1.5 group for both tissues (log2-fold change −0.41 and −0.45, respectively). The ENSSSCG00000022842, or *LOC100525692,* also was less expressed in the SOY1.5 group in both skeletal muscle (log2-fold change −1.8) and liver (log2-fold change −2.1), and the same occurred for *ALG6* (log2-fold change −0.68 and −0.74, respectively). Finally, the ENSSSCG00000051557 showed higher expression in the SOY1.5 group for both the skeletal muscle (log2-fold change +1.28) and liver (log2-fold change +1.36).

### 3.4. Functional Enrichment Analysis for Skeletal Muscle Differential Expression

Seven different pathway maps were detected (*p*-value < 0.10), which are linked to the Fatty Aldehyde Dehydrogenase or Aldehyde Dehydrogenase Family 3 Member A2 *(AL3A2),* Alpha-2-Glycoprotein 1, Zinc-Binding *(AZGP1)*, and T-Cell Surface Glycoprotein *(CD4)* genes (Table 4).

The DEG *AL3A2* was present in five (Appendix A) of the seven pathway maps identified, which encodes the AL3A2 enzyme. In this study, the *AL3A2* was less expressed in the SOY1.5 group (log2-fold change −0.77).

The *AZGP1* gene is involved in the *TNF-alpha* and *IL-1 beta (IL-1b)* pathways, which are involved with the occurrence of dyslipidemia and inflammation in adipocytes, leading to the development of obesity and type 2 diabetes diseases (Appendix A). Herein, the *AZGP1* gene was less expressed in the SOY1.5 group (log2-fold change −2.67). Thus, the increase in soybean caused an increase in AZGP1 expression. Another DEG identified in this study was *CD4* enriched in the “breakdown of CD4+ T cell peripheral tolerance in type 1 diabetes mellitus” pathway in muscle (Appendix A), which was less expressed in the SOY1.5 group (log2-fold change −1.57).

The enriched pathways show our DEG identified by DESeq2 with an important role in metabolism and diseases (Table 5). The majority of the top 10 enriched *(p*-value < 0.05) process networks identified are associated with immune response, with the same observed in the pathway maps. The identified networks, such as “immune response antigen presentation” with the DEG *AZGP1* and *CD4* (log2-fold change −2.67; log2-fold change −1.57), and “Kallikrein- kinin system” with the DEG *A2M* (alpha-2-macroglobulin; log2-fold change −1.79), are shown in Appendix A.

### 3.5. Functional Enrichment Analysis for Liver Differential Expression

Six different pathway maps were detected (*p*-value < 0.10), which are linked to BAG chaperone 1 *(BAG-1), ST13* Hsp70-Interacting Protein (Hip), Microtubule-Associated Protein Tau *(MAPT),* group of non-phosphorylatable alkali light chains of Myosin II *(MELC),* Protein Phosphatase 2 Catalytic *(PP2C),* Adenine nucleotide translocases Protein group *(ANT),* Peptidylprolyl Isomerase F *(PPIF),* and cyclin-dependent kinase inhibitor 1A *(CDKN1A or p21)* genes (Table 6).

Herein, we identified the DEG *BAG-1* as an up-regulated gene (log2-fold change +0.35) in liver samples from pigs fed 1.5% of soybean oil. This gene participates in the pathway map named HSP70 and HSP40-dependent folding in Huntington’s disease *BAG-1* (Appendix A).

We identified the *ST13* gene as a DEG in the liver of pigs that were fed with different levels of soybean oil (1.5% vs. 3%), being less expressed in the SOY1.5 group (log2-fold change −0.35). In the “SP70 and HSP40-dependent folding in Huntington’s disease” pathway, ST13 is related to the folding process of mutant Huntington via stimulation of HSP70 (Appendix A).

The microtubule-associated protein *Tau (MAPT)* gene was identified as DEG in the liver of our pig population, showing lower expression (log2-fold change −1.18) in the SOY1.5 group and enriched in “Inhibition of remyelination in multiple sclerosis: regulation of cytoskeleton proteins” (Appendix A) and “Tau pathology in Alzheimer disease” (Appendix A). Another interesting DEG, identified herein, also enriched in the “inhibition of remyelination in multiple sclerosis: regulation of cytoskeleton proteins” pathway, was the Myosin light chain 3 (*MYL3*). The *MYL3* participates in the pathway as a MELC group with lower expression (log2 fold change −1.37) in the SOY1.5 group.

The solute carrier family 25-member 4 *(SLC25A4)* gene was identified as DEG in the liver of pigs that were fed with different levels of soybean oil (1.5 vs. 3%), more expressed (log2-fold change +0.57) in the SOY1.5 group. The DEG *SLC25A5* was involved in the “mitochondrial dysfunction in neurodegenerative diseases” enriched pathway (Appendix A), as the adenine nucleotide translocases protein group (*ANT*).

Another gene identified as a DEG in the liver of pigs that were fed with different levels of soybean oil (1.5 vs. 3%) is the peptidylprolyl isomerase F (*PPIF*), a member of the peptidyl–prolyl cis-trans isomerase family (*PPIase*), that was more expressed (log2-fold change +0.81) in the SOY1.5 group. In the “mitochondrial dysfunction in neurodegenerative disease” pathway (Appendix A), the DEG *PPIF* is linked to the *ANT* gene. The top 10 enriched process networks (*p*-value < 0.10) show our DEG identified with an important role in processes such as “protein folding in normal condition” and “inflammation_Kallikrein-kinin system” (Figure 1).

The “protein folding in normal condition” network presents the DEG *PPID* (+0.48log2 fold change), HSP105 (+0.83log2 fold change), *BAG-1* (+0.35log2 fold change), and *ST13* (Hip) (−0.35log2 fold change) (Appendix A). The “inflammation_Kallikrein-kinin system” contains the DEG *KNG1* (−0.67log2 fold change), which in the gene network appears enriched as a receptor ligand and has bioactive metabolites of bradykinin, as can be seen in Appendix A.

## 4. Discussion

In this study, our main objective was to evaluate the effect of increasing the level of soybean oil present in swine feed on the transcriptomic profile of skeletal muscle and liver. These two target tissues were selected for this study due to their importance in lipid metabolism, hence their role in metabolic diseases [40,41,42]. These previous studies suggest that this skeletal muscle–liver signaling axis could be a target for therapies against obesity and other metabolic diseases.

This nutrigenomics study, in which the level of oil used in the diet was the main focus, corroborates with previous studies that suggested that the diet influences the transcriptome [43,44]. However, this is the first study that shows the effect of different level of dietary oil on a pig’s transcriptome. This dietary approach in pig production showed better values of feed efficiency, growth performance, and carcass characteristics according to Adeola and Bajjalieh [45], de Llata et al. [46] and Zhang et al. [47], which can be an interesting alternative for pig production.

### 4.1. Different Levels of Dietary Soybean Oil Modulates Fat Deposition

The increase in soybean oil in the pig diet influenced the fatty acid profile of skeletal muscle and liver. We observed higher deposition of MUFA with a higher level of soybean added in the diet (Table 1 and Table 2). On other hand, we identified the decrease in the deposition of total n-3 and n-6 in these tissues, which corroborates with the previous study by Alencar et al. [27].

In a study with *longissimus dorsi* muscle samples from pigs receiving a diet enriched with linseed (rich in AL), in the growing and finishing [48] identified genes annotated as involved in apoptosis, muscle organ development, and transcriptional regulation, the PUFA n-3 presented a fundamental role in the development and maintenance in the healthy animal and its ingestion is related to action effects of insulin, neurologic development, reproduction, innate and acquired immunity, and with transcription factor expression [48]. Beyond these, genes involved in the metabolic process of glycose, amino acid metabolism, the quinase IκB cascade *NF-κB*, FA metabolic process, IGF-1/insulin and of genes that code for Wnt pathway elements are also involved. As in Alencar et al. [27], regardless of the amount of soybean oil adopted, the values were above the recommended (ratio <4:1). Values of the n-6:n-3 ratio above the recommended are related to pro-inflammatory immune responses and metabolic diseases.

The SFA and MUFA in the pigs have in vivo synthesis with low influence by diet [49], which can be observed in our results in which SFA did not present changes with the modification between oil, such as the MUFA C20:1. Regarding MUFA, we identified a difference in both intramuscular fat and liver; the diet is responsible for a considerable amount of OA in the adipose tissue. The de novo synthesis is the main one in the supply of MUFA to the body, so the tissue content is not so influenced by the diet.

Most serum parameters showed no difference with the addition of soybean oil to the diet as shown in our previous study [31]. With the increase in oil, we observed lower values of albumin, triglycerides, and VLDL. In the DEG identified in Appendix A, we observed APOO with higher expression (log2-fold change +0.60) in the SOY1.5-encoded protein associated with HDL, LDL, and VLDL lipoproteins [39]; alterations in members of the apolipoprotein family may result in changes in the regulation of particle metabolism, as with VLDL, which may be related to the higher VLDL content found in SOY1.5.

### 4.2. Different Levels of Dietary Soybean Oil Modulate Gene Expression in Skeletal Muscle

The *AL3A2* gene is present in most enriched pathway maps, both in normal and pathologically related pathways. The AL3A2 enzyme helps to detoxify aldehydes produced by alcohol metabolism and lipid peroxidation, the development of the central and peripheral nervous systems, and the oxidation of long-chain aliphatic aldehydes to fatty acid [39]. A previous study demonstrated that the *AL3A2* gene participates in the oxidation of 12-oxo-dodecanoic acid to dodecanedioic acid through the activation effect by the catalysis mechanism [50]. This gene also participates in the “leukotriene 4 biosynthesis and metabolism” pathway, where the Leukotriene E4 is formed by a family of inflammatory lipid mediators and synthesized from arachidonic acid through a diversity of cells, such as basophils, mast cells, eosinophils, and macrophages, being the final step of inactivation in “leukotriene B4 metabolism” [51]. Furthermore, the *AL3A2* gene is involved in “triacylglycerol metabolism” and the “oxidative stress in adipocyte dysfunction in type 2 diabetes and metabolic syndrome X” pathway, in which the accumulation of oxidative products occurs due to an imbalance in the net levels of reactive oxygen species (ROS) in relation to the antioxidant capacity of the body [52]. The different levels of soybean oil in the diet could be affecting the *AL3A2* expression level. With the increase in soybean oil in the diet, an increase in *AL3A2* occurs. Thus, the consumption of higher levels of soybean oil could improve the rate of lipid oxidation, accumulation of free radicals into the cells causing cell damage and, consequently, the inflammatory condition that would lead to metabolic diseases such as type 2 diabetes and atherosclerosis [53]. These results confirm previous studies that identified the fatty aldehyde dehydrogenase isozymes, such as *AL3A2,* have cell-specific functions associated with inflammation, differentiation, or oxidative-stress responses [54,55].

The DEG *AZGP1* as observed increases with increasing dietary oil inclusion; is related to stimulation of the breakdown of lipids within adipocytes, which causes fat to be lost in some advanced cancers; and is also capable of binding to polyunsaturated fatty acids [56]. Studies have shown that the *AZGP1* gene can be inhibited by *TNF-alpha* and other genes related to the development of metabolic disorders. Also, this gene encodes a soluble protein classified as an adipokine [57,58]. In an obese adipose tissue, the overexpression of *TNF-alpha* together with IL-1b activates the ERK1/2 pathways leading to impaired gene expression that are involved in inflammation, FA oxidation, lipolysis, lipogenesis, in addition to oxidative stress [59]. Our studies corroborate with previous nutrigenomics studies which showed that the *AZGP1* expression level can be affected by the diet [60,61]. The *AZGP1* gene plays a fundamental role in lipid metabolism and other metabolic diseases such as cancer, being considered as a lipid-metabolizing factor [62] that impacts the fatty acid metabolism, increasing the lipolysis process and decreasing the inflammation signs [63,64]. According to our findings, the AZGP1 gene-enriched pathway in the skeletal muscle is directly related to dyslipidemia, inflammatory response mediated by TNF-alpha and IL-1b, and metabolic diseases.

The *CD4* gene is a down-regulated DEG in the skeletal muscle of pigs fed with a lower level of soybean oil (SOY1.5). As well as facilitating T-cell activation, *CD4* is an important mediator of indirect neuronal damage in infections and immune-mediated conditions affecting the nervous system. Studies have shown that diet-induced obesity increases the expression of T-cell and MHC II molecules in adipose tissue and fat spots in this tissue [65]. Skeletal muscle is an important organ for insulin response; thus, it presents a significant contribution in systemic insulin sensitivity [66]. Hong et al. [67] suggested that the obesity condition affects the inflammatory state of skeletal muscle. Corroborating, Varma et al. [68] showed that diets with a high level of fat can induce obesity inflammation by the macrophage infiltration into the skeletal muscle. However, dietary intake of specific PUFAs can modulate the inflammatory responses that can be explained by the influence of the dietary fatty acid profile into the n-6:n-3 PUFA ratio of the membrane phospholipids. The n-6 PUFA metabolism generates lipid mediators that have pro-inflammatory functions while n-3 has an anti-inflammatory effect [69,70,71]. Therefore, the overexpression of lipid mediators generated by n-6 are associated with inflammatory diseases [69,70,71]. These previous findings corroborate with our results.

### 4.3. Soybean Oil Added to Pig’s Diet Modulates Gene Expression in Liver Tissue

The results obtained through the enriched pathways are related to the importance of the liver as a central organ for systemic metabolism [72]. Some pathways have been enriched and are important signaling pathways associated with disease, that help to understand the mechanism of genes altered by diet. The liver plays a fundamental role in the energy balance of the entire body, removes toxins such as ammonia, and is responsible for the detoxification of most endogenous and exogenous toxic compounds. When some hepatic mechanism fails, it causes problems that can reach the brain, affecting brain function and causing several neurodegenerative diseases including Huntington’s and Alzheimer (HD and AD) [73,74].

In HD, the majority of symptoms are related to neuronal damage although additional peripheral tissue abnormalities such as energy metabolism deficiency, skeletal muscle atrophy, and adipose tissue dysfunction, which have been reported in both humans and mice with HD [75]. This relationship may explain the findings in the liver of protein-coding genes with functions associated with neuronal balance. In mice, the aggregates of the resultant mutant Htt protein (mHtt) were found from the transgenic mouse model; mHtt is related with the suppression of the transcription factor *C/EBPα* (TF critical for energy homeostasis) and the *PPARγ* protein function. There is evidence that defects in liver function may contribute to peripheral abnormalities in HD mice [75].

The *BAG-1,* which has increased expression in the SOY1.5 group, is a protein-coding gene which is involved in binding to the membrane protein *BCL2,* which participates in the regulation of apoptosis, oncogenesis, neuronal differentiation, and the reactions of cellular-regulatory proteins, including glucocorticoid receptor [76]. *BAG-1,* when associated with *BCL2,* activates the degradation of the Huntingtin mutation, probably via activation of U-box containing protein 1 (CHIP) and proteasome [76]. Members of the *BAG* and *CHIP* family may also play an important role in modulating neurodegeneration by targeting misfolded mutant proteins to the ubiquitin–proteasome system [77]. *BAG-1* is a co-chaperone for HSP70 and HSC70 chaperone proteins, that participates as a nucleotide exchange factor (NEF) promoting ADP release from HSP70 and HSC70 proteins [39]. The HPS70 plays a protective role in several different models of nervous system injury, although it has an association with a deleterious role in some diseases [78]. Pathologies related to misfolded proteins affect different classes of neurons and are related to numerous diseases such as Parkinson disease, AD, amyotrophic lateral sclerosis (ALS), and inheritable polyglutamine (PolyQ) diseases [78]. In the study by Nollen et al. [79] with mammalian tissue culture cell lines, *BAG-1* participated as an inhibitor of Hsp70-dependent refolding; the authors showed that a two-fold increase in cellular levels of *BAG-1* can inhibit Hsp70 refolding. Adrie et al. [80] showed that *BAG-1, BAG-3,* cellular inhibitor of apoptosis 1 (cIAP1), and myeloid cell leukemia sequence 1 (MCL-1) were more expressed in brain-dead patients than in hip surgery patients, demonstrating an inhibition of the apoptosis process. The study by Sroka et al. [76] demonstrated that BAG-1 plays a key role in neuronal survival and differentiation, and is essential for the proper development and maintenance of the central nervous system. In disease-related pathways, BAG-1 has its function compromised and this causes BAG-1 accumulation, as in the enriched pathway “HSP70 and HSP40-dependent folding in Huntington’s disease”, which in the SOY1.5 group with higher expression would lead to even more ubiquitin–proteasome activation; however, with HD disease, proteasome functionality is impaired and the impairment of proteasome function leads to changes in neurotransmitter systems [81].

On the other hand, the ST13 or Hip co-chaperone showed lower expression in the SOY1.5 group, the gene encoding the HSP70 interaction protein, which can also collaborate with other positive cofactors, such as the organizer protein HSP70-HSP90 (Hop), or compete with negative cofactors such as *BAG-1* [82]. The *ST13* can also act as a facilitator of the Hsc/HSP70 chaperone, aiding in protein folding and repair, and controlling the activity of proteins responsible for regulation, such as steroid receptors and regulators of proliferation or apoptosis [82]. According to Hou et al. [83], in case of functional alteration of the *ST13* gene, the HSP70 protein can lose control over the apoptosis process, and thus generate an incorrect proliferation. Freitas et al. [84], studying miRNA and mRNA expression in peripheral blood cells, found miR-107 down-regulated, and its target, the *ST13* mRNA, showed high expression in patients with high platelet reactivity. miR-107 is associated with expression inhibition of genes involved in metabolism, cell division, angiogenesis, and stress response [85], and has been associated with neurodegenerative diseases and cancer [85]. In the SOY1.5 group, the expression was lower, decreasing its ability to prevent the newly formed Hsc/Hsp70-substrate complex from prematurely dissociating in the “HSP70 and HSP40-dependent folding in Huntington’s disease” pathway.

The *Tau/MAPT* was also enriched in the “Tau pathology in Alzheimer disease” pathway together with protein phosphatase; PPM1F was identified in the pathway as the PP2C group with the highest expression in the SOY1.5 group. *PPM1F* phosphatase dephosphorylates and negatively regulates the activities of MAP kinases, in addition to being able to interact with guanine Rho nucleotide exchange factors [86]. The AD is characterized by the extracellular accumulation of amyloid beta peptides or plaques (Aβ) and the intracellular accumulation of phosphorylated tau protein species (tau tangles) in the brain. The Hsp-70/Hsc-70 chaperone, mentioned above, can bind to the *Tau/MAPT* protein, reducing its phosphorylation and inducing its proteasomal degradation. In addition, Hsp-70 is involved in mediating the degradation of Tau/MAPT by recruiting the CHIP E3 ubiquitin ligase, whereas *BAG-1* can associate with *Tau/MAPT* in a Hsc-70-dependent manner and inhibit its proteasomal degradation [87,88]. In the study by Elliott; Laufer; Ginzburg, [88] in AD hippocampi, the authors reported that *BAG-1* co-localizes with both tau tangles and intracellular amyloid which may indicate that BAG-1 plays a significant role in AD pathology.

As Tau (MAPT) are the proteins that stabilize microtubules, the SOY1.5 group was less expressed, thus potentiating the non-stabilization of microtubules, which is harmful. Duan et al. [89] suggested that the abnormal hyperphosphorylation of *Tau/MAPT* is caused by the conformational change of this protein in the diseased brain, making this a favorable environment for phosphorylation or an unfavorable one for dephosphorylation. Neurodegenerative disease such as AD is characterized by the microtubule-associated neuronal protein *MAPT* that undergoes hyperphosphorylation by multiple kinases resulting in microtubule disintegration, and this phosphorylation can be regulated by several phosphatases, including *PP2C* [86]. In the study by Liu et al. [86] with several pools of Tau proteins isolated from an AD brain, it was observed that Tau is abnormally hyperphosphorylated and glycosylated when there is an imbalance between phosphorylation and dephosphorylation, favoring the formation of intraneuronal neurofibrillary tangles (NFTs) which is one of the histopathology’s characteristics of AD. Thus, the authors suggested that tau glycosylation is an early abnormality that facilitates hyperphosphorylation in the AD brain [86]. In the “inhibition of remyelination in multiple sclerosis: regulation of cytoskeleton proteins”, *MAPT*—in which Fyn suppression inactivates the guanine nucleotide exchange factor Vav 2 via *CDC42*—destabilizes the actin microfilaments. Fyn suppresses glucocorticoid receptor DNA-binding factor 1, leading to RhoA activation. This suppression of Fyn also impairs the interaction with *Tau (MAPT)* and alpha tubulin [90]. In the “inhibition of remyelination in multiple sclerosis: regulation of cytoskeleton proteins”, RhoA binds to protein kinase 2 (ROCK2), which contains association with Rho and promotes phosphorylation and activation of protein phosphatase 1 (MLCP (reg))/protein phosphatase 1, isoenzyme beta (MLCP) (cat)), with dephosphorylation of Myosin II regulatory light chains (MRLC and MELC) [91,92]. This activation of RhoA can lead to a growth-related inhibition of the oligodendrocyte process and impair remyelination in multiple sclerosis [39].

The neurodegenerative diseases can be involved in the intracellular Ca2+ through losing by both homeostasis and excitotoxicity [93]. An overload of intracellular Ca2+ high concentration reflects in an overload of the UCP2 and UCP3 pathway in the mitochondrial Ca2+, which can induce the depolarization of the mitochondrial membrane and, as one of the consequences, activate the production of reactive oxygen- species (ROS) and stimulate the MPTP complex by the Cardiolipin/ANT pathways in PPIF [94,95].

The expression of *p21* is controlled by the *p53* tumor-suppressor protein that, in response to stress stimuli, mediates the *p53*-dependent cell cycle G1 phase [39]. The Huntingtin mutation can either activate or decrease *p53* activity, and an inhibition of p53 in the “mitochondrial dysfunction in neurodegenerative diseases” pathway can lead to *p21* under-expression, leading to the inhibition of neuronal cell survival [96]. The gene expression analysis performed in transgenic mice with fatty livers in the study by Yahagi et al. [97] resulted in increased expression of *p21* mRNA, considered an indicator of *p53* activity. Thus, the cellular toxicity of excess FFA, lipid peroxidation, and oxidative stress may be associated with the mechanisms of activation of the *p53* protein [89].

Bradykinin plays an important role in mediating inflammation resulting in vasodilation, in addition to stimulating prostaglandin synthesis [98]. Process networks were related to both the chaperone group (HSP90) and pro-inflammatory activity, respectively. These results from the gene networks corroborate with the pathway maps shown above, highlighting that the change in the amount of oil in the diet of male pigs causes a change in the gene expression.

### 4.4. Genes Common to Dietary Treatments and Overview of the Effect of Soybean Oil Addition in Different Tissues

By identifying the common genes between the tissues, we observed the influence on modulation and important relationship to cell-cycle-related kinase, in addition to the relationship to the Ca, cAMP, and lipid-signaling pathways and glycosylation.

As observed and discussed regarding the genes that showed differential expression, we observed a better relationship when using SOY3.0 associated with the enriched pathways presented and the association with gene function. In liver tissue, we identified 45 DEG and its relation to neurodegenerative diseases, which highlights the importance of the liver in disease regulation. In contrast, in skeletal muscle, 281 DEG were identified with greater relation to FA metabolism, metabolic diseases, and inflammatory processes. Our findings show similar function-signaling pathways, process networks, and DEG with respect to the changes that were obtained by modulating the FA profile, making an important contribution to nutrigenomics studies.

## 5. Conclusions

In this nutrigenomics study, we verified that increasing the level of soybean oil in the diet of pigs, an animal model for metabolic diseases in humans, affected the transcriptome profile of skeletal muscle and liver tissue. The differentially expressed genes identified here were related to relevant biological processes and participated in process networks and pathway maps associated with metabolic and neurodegenerative diseases, such as “TNF-alpha, IL-1 beta induces dyslipidemia and inflammation in obesity and type 2 diabetes in adipocytes” and “HSP70 and HSP40-dependent folding in Huntington’s disease”. The higher level of soybean oil added to the diet led to a higher expression of genes targeting biological processes related to lipid oxidation and consequently to metabolic diseases and inflammation. These findings may help us to better understand the biological mechanisms that can be modulated through the diet, specifically—in this study—by increasing the level of soybean oil (an important source of unsaturated fatty acids) in the diet of pigs.

## Figures and Tables

**Figure 1 animals-12-01632-f001:**
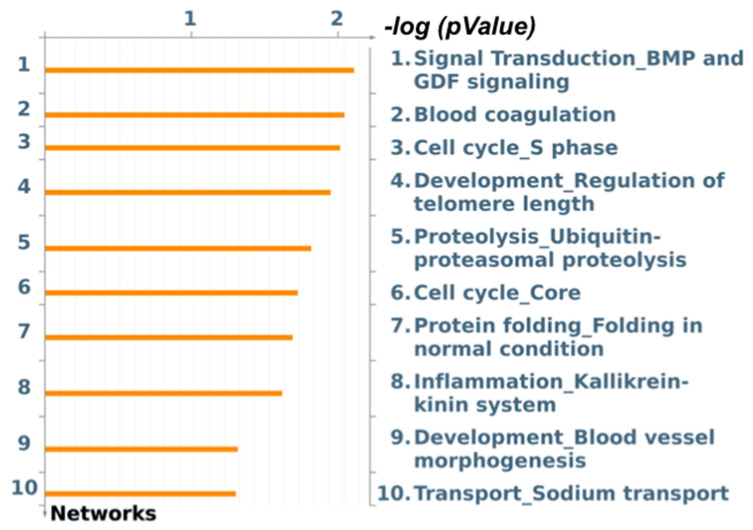
Top 10 enriched process networks identified by MetaCore software from the list of differentially expressed genes (FDR 10%) in the liver of pigs fed with two different levels of soybean oil in the diet (1.5% and 3.0% of soybean oil).

**Table 1 animals-12-01632-t001:** Effects of dietary treatments on fatty acid profile of *Longissimus lumborum* intramuscular fat of pigs.

Fatty Acid (%)		Dietary Treatment		
	SOY1.5	SOY3.0	Pooled SEM ^1^	*p*-Value
Saturated fatty acid (SFA)				
Myristic acid (C14:0)	1.14	1.19	0.04	0.20
Palmitic acid (C16:0)	25.50	25.01	0.21	0.21
Stearic acid (C18:0)	12.18	11.89	0.15	0.42
Monounsaturated fatty acid (MUFA)				
Palmitoleic acid (C16:1)	2.86	3.17	0.13	0.02
Eicosenoic acid (C20:1)	0.51	0.55	0.03	0.11
Oleic acid (C18:1 n-9)	38.93	44.15	1.40	<0.01
Polyunsaturated fatty acid (PUFA)				
Linoleic acid (C18:2 n-6)	17.90	13.28	1.12	<0.01
Alpha-linolenic acid (C18:3 n-3)	0.77	0.56	0.06	<0.01
Eicosapentaenoic acid (C20:5 n-3)	0.30	0.15	0.09	0.12
Docosahexaenoic acid (C22:6 n-3)	0.36	0.16	0.08	0.03
Total SFA	38.83	38.09	0.65	0.26
Total MUFA	42.29	47.70	1.48	<0.01
Total PUFA	19.28	14.80	1.72	0.02
Total n-3 PUFA ^2^	1.35	0.87	0.15	<0.01
Total n-6 PUFA ^3^	17.90	13.28	1.12	<0.01
PUFA:SFA ratio ^4^	0.50	0.39	0.05	0.03
n-6:n-3 PUFA ratio ^5^	14.20	17.29	1.70	0.10
Atherogenic index	0.49	0.48	0.09	0.43

Pigs (*n* = 36) were fed either a corn–soybean meal diet containing 1.5% soybean oil (SOY1.5) or diet containing 3% soybean oil (SOY3.0). Values represent the least square means from a subset of pigs (*n* = 36; 18 pigs/treatment). ^1^ SEM = standard error of the least square means. ^2^ Total n-3 PUFA = {[C18:3 n-3] + [C20:5 n-3] + [C22:6 n-3]}. ^3^ Total n-6 PUFA = C18:2 n-6. ^4^ PUFA:SFA ratio = total PUFA/total SFA. ^5^ Σ n-6/Σ n-3 PUFA ratio.

**Table 2 animals-12-01632-t002:** Effects of dietary treatments on fatty acid profile of liver tissue of pigs.

Fatty Acid (%)	Dietary Treatment		Pooled SEM ^1^	*p*-Value
	SOY1.5	SOY3.0		
Saturated fatty acid (SFA)				
Myristic acid (C14:0)	0.73	0.98	0.05	<0.01
Palmitic acid (C16:0)	20.92	22.98	0.40	<0.01
Stearic acid (C18:0)	25.48	21.28	1.06	<0.01
Monounsaturated fatty acid (MUFA)				
Palmitoleic acid (C16:1)	0.66	0.93	0.05	<0.01
Oleic acid (C18:1 n-9)	21.36	27.84	1.06	<0.01
Polyunsaturated fatty acid (PUFA)				
Linoleic acid (C18:2 n-6)	27.02	23.64	0.67	<0.01
Alpha-linolenic acid (C18:3 n-3)	1.42	1.17	0.10	0.07
Eicosapentaenoic acid (C20:5 n-3, EPA)	0.58	0.27	0.11	0.04
Docosahexaenoic acid (C22:6 n-3, DHA)	1.18	0.99	0.11	0.17
Total SFA	46.69	45.24	1.03	0.31
Total MUFA	22.01	28.78	1.04	<0.01
Total PUFA	30.79	26.06	0.55	<0.01
Total n-3 PUFA ^2^	3.75	2.42	0.37	<0.01
Total n-6 PUFA ^3^	27.02	23.64	0.67	<0.01
PUFA:SFA ratio ^4^	0.67	0.58	0.02	<0.01
n-6:n-3 PUFA ratio ^5^	8.51	9.90	0.50	0.05
Atherogenic index	0.42	0.51	0.01	<0.01

Pigs (*n* = 35) were fed either a corn–soybean meal diet containing 1.5% soybean oil (SOY1.5) or diet containing 3% soybean oil (SOY3.0). Values represent the least square means from a subset of pigs (*n* = 35; 17 pigs/SOY1.5; 18 pigs/SOY3.0). ^1^ SEM = standard error of the least square means. ^2^ Total n-3 PUFA = {[C18:3 n-3] + [C20:5 n-3] + [C22:6 n-3]}. ^3^ Total n-6 PUFA = C18:2 n-6. ^4^ PUFA:SFA ratio = total PUFA/total SFA. ^5^ Σ n-6/Σ n-3 PUFA ratio.

**Table 3 animals-12-01632-t003:** Common differentially expressed genes between the two tissues comparisons, in the skeletal muscle and liver tissue of pigs fed with two different levels of soybean oil in the diet (1.5% and 3.0% of soybean oil).

Gene Common	Description	Reference
ENSSSCG00000009578 Cyclin-dependent kinase 20 (*CDK20*)	Cell-cycle-related kinase. Its expression is related to the activation of β-catenin-TCF signaling and cell cycle progression. Can activate cyclin-dependent kinase 2 which is related to cell growth.	[38,39]
ENSSSCG00000014903 Coiled-coil domain-containing 90B (*CCDC90B*)	Paralog of the MCUR1 gene (Mitochondrial Calcium Uniporter Regulator 1) which is related to the Ca, cAMP, and lipid-signaling pathways.	[39]
ENSSSCG00000022842 LOC100525692	Protein-encoding gene.	[39]
ENSSSCG00000022842 Alpha-1,3-Glucosyltransferase (*ALG6*)	Related to N-linked glycosylation.	[39]
ENSSSCG00000017914 Glycolipid Transfer Protein Domain-Containing Protein 2 *GLTPD2*	Participates in the transfer of glycolipids.	[39]
ENSSSCG00000051557	-	-

**Table 4 animals-12-01632-t004:** Pathway maps identified by MetaCore software (*p*-value < 0.10) from the list of differentially expressed genes (FDR 10%) in the skeletal muscle of pigs fed with two different levels of soybean oil in the diet.

Pathway Maps	*p*-Value	DEG ¹
Fatty acid omega oxidation	0.0333	*AL3A2*
Leukotriene 4 biosynthesis and metabolism	0.0442	*AL3A2*
*TNF-alpha, IL-1* beta induces dyslipidemia and inflammation in obesity and type 2 diabetes in adipocytes	0.0464	*AZGP1*
Breakdown of CD4+ T cell peripheral tolerance in type 1 diabetes mellitus	0.0539	*CD4*
Triacylglycerol metabolism p.1	0.0656	*AL3A2*
Oxidative stress in adipocyte dysfunction in type 2 diabetes and metabolic syndrome X	0.0699	*AL3A2*
Peroxisomal branched-chain fatty acid oxidation	0.0908	*AL3A2*

¹ Differentially expressed genes (DEG).

**Table 5 animals-12-01632-t005:** Process Networks identified by MetaCore software (*p*-value < 0.10) from the list of differentially expressed genes (FDR 10%) in the skeletal muscle of pigs fed with two different levels of soybean oil in the diet.

Process Networks	*p*-Value	DEG ¹
Chemostaxis	0.0018	*CCR10, GPCRs, CD4*
Cell adhesion_Leucocyte chemostaxis	0.00378	*CCR10*,* GPCRs*,* CD4*
Immune response_Antigen presentation	0.0046	*CD4*,* AZGP1*
Signal transduction_Leptin signaling	0.0156	*A2M*,* T-A2MG*
Inflammation_Kallikrein–kinin system	0.0443	*A2M*,* T-A2MG*
Reproduction_Male sex differentiation	0.0699	*Tektin 1*,* AKAP3*

¹ Differentially expressed genes (DEG).

**Table 6 animals-12-01632-t006:** Pathway maps with DEG between SOY1.5 vs. SOY3.0 in liver tissue enriched in significant pathways (*p*-value < 0.10).

Pathway Maps	*p*-Value	DEG ¹
HSP70 and HSP40-dependent folding in Huntington’s disease	0.01034	*BAG-1*,* ST13 (Hip)*
Inhibition of remyelination in multiple sclerosis: regulation of cytoskeleton proteins	0.03022	*MAPT*,* MELC*
Tau pathology in Alzheimer disease	0.04543	*MAPT*,* PP2C*
Mitochondrial dysfunction in neurodegenerative diseases	0.05153	*ANT*
Dual role of *p53* in transcription deregulation in Huntington’s Disease	0.07179	*p21*
LRRK2 in neuronal apoptosis in Parkinson’s disease	0.09869	*ANT*

¹ Differentially expressed genes (DEG).

## Data Availability

The dataset supporting the conclusions of this article is available in the in the European Nucleotide Archive (ENA) repository (EMBL-EBI), under accession PRJEB50513 and PRJEB52629 [http://www.ebi.ac.uk/ena/data/view/, (accessed on 12 December 2021), PRJEB50513; http://www.ebi.ac.uk/ena/data/view/, (accessed on 12 December 2021), PRJEB52629].

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
