# Peer review of "Differential Gene Expression Associated with Soybean Oil Level in the Diet of Pigs"

_animals, 2022, doi:10.3390/ani12131632_

Round 1

Reviewer 1 Report

line 40: may verified …as biomedical model

line 67: dot is missing at the end of that sentence

line 82: Sterol regulatory-element binding proteins can be written with lowercase initial

line 82: Nuclear factor kappa B can be written with lowercase initial

line 87: delete round brackets

line 97: the literature reference number is missing

line 143: “genetically lean”, word “purebred” is proper

line 146: meaning of sentence is not clear, were the individuals included in the study tested free of genetic defect?

line 146: the test used to verify the genotype is not clear, has direct gene test for mutations in the ryanodine receptor 1 (RyR1) or indirect volatile anesthetic halothane gas test been performed?

line 148: it is necessary to mention that the study fit into the course of commercial fattening, if so.

line 153: write fattening day

line 154: write fattening day

line 165: at what age was the slaughtering made, at what age were the tissue samples collected?

line 165: comma before literature reference number is to delete

line 166: comma before literature reference number is to delete

line 169: comma before year is to delete

line 219: space between author’s names is to delete

line 267: “e”?

line 269: “e”?

line 476: please, detail the own differences identified between the diets

I suggest rewording the Conclusions so that the authors give not only the existence of relationships but also their direction.

Take a stand on which soybean oil supplement is more affordable.

Do the authors consider it possible that the results obtained could have been influenced by immunocastration as a method?

Author Response

Piracicaba, São Paulo State, Brazil, June 5, 2022.

Dear Reviewer #1,

      We are returning the revised manuscript entitled “Differential gene expression associated with soybean oil level in the diet of pigs” (Submission ID: 1734763) to Animals for review and possible publication. We sincerely appreciate your comments/suggestions, which were very helpful to improve the quality of our manuscript. Please see below the point-by-point response and the edits in the manuscript were highlighted in yellow.

Reviewer: line 40: may verified …as biomedical model

Authors: We changed biomedical model to animal model. (Lines 44, 52, 73, and 75)

 Reviewer: line 67: dot is missing at the end of that sentence

Authors: The correction was made. (Line 68)

 Reviewer: line 82: Sterol regulatory-element binding proteins can be written with lowercase initial

Authors: The correction was made. (Line 83)

 Reviewer: line 82: Nuclear factor kappa B can be written with lowercase initial

Authors: The correction was made. (Line 84)

 Reviewer: line 87: delete round brackets

Authors: The correction was made. (Line 99)

 Reviewer: line 97: the literature reference number is missing

Authors: The correction was made. (Line 99)

 Reviewer: line 143: “genetically lean”, word “purebred” is proper

Authors:  The correction was made. (Line 146)

 Reviewer: line 146: meaning of sentence is not clear, were the individuals included in the study tested free of genetic defect?

Authors: The test performed to evaluate the halothane-free genotype was molecular test by PCR-RFLP according to Fujii, et al. 1991. This information was added in the text. (Lines 148 and 149)

 Reviewer: line 148: it is necessary to mention that the study fit into the course of commercial fattening, if so.

Authors: The study was performed at an experimental farm of DB Genética company. This information was added in the text. (Lines 157 and 158)

Reviewer: line 153: write fattening day

Authors:  The replacement was made. (Line 155)

 Reviewer: line 154: write fattening day

Authors:  The replacement was made. (Line 156)

Reviewer: line 165: at what age was the slaughtering made, at what age were the tissue samples collected?

Authors: This information was added in the text. (Line 158)

Reviewer: line 165: comma before literature reference number is to delete

Authors: The correction was made. (Line 168)

 Reviewer: line 166: comma before literature reference number is to delete

Authors: The correction was made. (Line 169)

 Reviewer: line 169: comma before year is to delete

Authors: The correction was made. (Line 172)

 Reviewer: line 219: space between author’s names is to delete

Authors: The correction was made. (Line 224)

Reviewer: line 267: “e”?

Authors: The correction was made. (Line 272)

Reviewer: line 269: “e”?

Authors: The correction was made. (Line 274)

Reviewer: line 476: please, detail the own differences identified between the diets

Authors: The statement related to the differences between diets that we observed in our study was added in the text. (Lines 394 – 398)

 Reviewer: I suggest rewording the Conclusions so that the authors give not only the existence of relationships but also their direction. Take a stand on which soybean oil supplement is more affordable.

Authors: We agree with the reviewer, then the conclusion was rewritten. (Lines 631 - 633)

 Reviewer: Do the authors consider it possible that the results obtained could have been influenced by immunocastration as a method?

Authors: We understand the reviewer’s concern, but given the consumer requirements regarding worldwide animal welfare, the surgical castration of piglets without pain, has become a standard procedure in pork production. So, immunocastration a more welfare-friendly alternative management is the overall practice in Brazilian pig industry. Those whom still produce surgical castration pigs are adapting to immunocastration system. We agree with the reviewer that probably the castration system could affect the transcriptome profile of the different tissues of the animals. However, in this study it was not the main goal once we followed the current castration system used by pig production.

Reviewer 2 Report

Major suggestions

1.       Without the ingredient and calculated or analysed nutrient composition of the experimental diets for pigs, it is difficult to judge the claim of the authors that this is a “nutrigenomics” study. 

2.       The authors have rightly pointed out the role of adipose tissue in fatty acid metabolism (FA) (line 80). Given the central role of adipose tissue in pig FA, I wonder why the authors did not perform transcriptome profiling of adipose tissue.

3.       Suggest taking Figure 1-8, 10-11 as a supplemental figure. Provided that the tissue samples or the extracted RNA samples are preserved, I strongly recommend that the authors confirm the expression pattern of the differentially expressed genes by qPCR. The qPCR results can be included in a table that can be presented in the manuscript.

4.       Line 218 & 222: What is the reason for using FDR < 0.10 and not FDR < 0.05? I suggest the authors present the reasons, the authors' impression of using such an FDR, and its implications for data interpretation in the manuscript, perhaps in the Discussion section.

5.       Line 707-709: Suggest to down tune the conclusion considering the above mentioned comments i.e. 1 to 4. 

Minor suggestions:

1.       Line 24: Expand MUFA and PUFA

2.       Line 24-27: Please rephrase the sentences. I suggest the authors to link these results, which were influenced by the treatments, especially to the physiology or biological mechanisms in pigs.

3.       Line 73, 97: Adopt the references according to the citation style of the journal.

4.       Line 110: Do the authors means “consumption of MUFA and PUFA by human….”? Rephrase.

5.       Line 451: Figure 9 is not clear. DEG can be either upregulated or downregulated compared with a treatment group; the present figure does not depict the pattern of gene expression.

6.       Line 476-479: Not clear what the authors wanted to state. Diet composition and results of growth performance and carcass traits are not shown in this manuscript.

Author Response

Piracicaba, São Paulo State, Brazil, June 5, 2022.

Dear Reviewer #2,

      We are returning the revised manuscript entitled “Differential gene expression associated with soybean oil level in the diet of pigs” (Submission ID: 1734763) to Animals for review and possible publication. We sincerely appreciate your comments/suggestions, which were very helpful to improve the quality of our manuscript. Please see below the point-by-point response and the edits in the manuscript were highlighted in yellow.

Reviewer: Without the ingredient and calculated or analysed nutrient composition of the experimental diets for pigs, it is difficult to judge the claim of the authors that this is a “nutrigenomics” study. 

Authors: We understand the reviewer concern. All the information of the ingredient, metabolic energy and fatty acid profile of the diets used during the different phases of the experiment was already published. We clarify this information in the text. (Lines 166)

 Reviewer: The authors have rightly pointed out the role of adipose tissue in fatty acid metabolism (FA) (line 80). Given the central role of adipose tissue in pig FA, I wonder why the authors did not perform transcriptome profiling of adipose tissue.

Authors’ Response: We agree with the reviewer that it would be very interesting to be able to study the adipose tissue transcriptome. However, the experimental design performed at the beginning of the project did not predicts this type of data collection, so our budget did not allow us to include this tissue in our study.

 Reviewer: Suggest taking Figure 1-8, 10-11 as a supplemental figure.

Authors’ Response: We agree, the figures were moved to the supplemental file.

 Reviewer: Provided that the tissue samples or the extracted RNA samples are preserved, I strongly recommend that the authors confirm the expression pattern of the differentially expressed genes by qPCR. The qPCR results can be included in a table that can be presented in the manuscript.

Authors: In the supplementary table S1, we added the information of the RNA integrity score (RIN) of all samples used in this study. All samples presented a RIN above of the minimum value recommended (RIN > 7,0). We understand the reviewer comment about qPCR test, however the integrity of the RNA extract from the samples was higher and according to previous studies this validation seems appropriate for methods that use microarrays or other specific probes, as they have higher technical variation and less replicability than Illumina RNA-Seq (Marioni et al., 2008). The depth of the RNASeq data generated in this study is also above of the recommended for differential expression analysis by previous studies (Sims et al., 2014; doi:10.1038/nrg3642). So, it was considered in our experimental design and budget proposal, once in our previous studies we verified a higher correlation between RNASeq and qPCR results (Oliveira et al., 2016; doi: 10.2527/jam2016-0338; Cesar et al., 2016; doi:10.1186/s12864-016-3306-x.

Reviewer: Line 218 & 222: What is the reason for using FDR < 0.10 and not FDR < 0.05? I suggest the authors present the reasons, the authors' impression of using such an FDR, and its implications for data interpretation in the manuscript, perhaps in the Discussion section.

Authors: We understand the reviewer's concern. The FDR threshold applied in this study was performed at the beginning of the project’s experimental design definitions, which was chosen according to previous studies including our previous publications (Cesar et al., 2015; Cesar et al., 2016), MDPI publications (Glotov et al., 2022; doi:doi.org/10.3390/genes13040574; Pareek et al., 2019); doi: 10.3390/vetsci6020036), and DESeq2 R package recommendation, which FDR cutoff of  > 0.1 is a default option for differential expression analysis. (Line 223)

Reviewer: Line 707-709: Suggest to down tune the conclusion considering the above mentioned comments i.e. 1 to 4.

Authors: We agree with the reviewer. The conclusion was rewritten. 

Reviewer: Line 24: Expand MUFA and PUFA

Authors: The words were expanded. (Line 24)

Reviewer:.Line 24-27: Please rephrase the sentences. I suggest the authors to link these results, which were influenced by the treatments, especially to the physiology or biological mechanisms in pigs.

Authors:  The sentence was rewritten as reviewer’s suggestion. (Lines 26 -28)

 Reviewer:  Line 73, 97: Adopt the references according to the citation style of the journal.

Authors: The correction was made. (Lines 74, 99)

Reviewer: Line 110: Do the authors means “consumption of MUFA and PUFA by human….”? Rephrase.

Authors: We agree with the reviewer and the phrase was rewritten. (Lines 111 - 114)

Reviewer: Line 451: Figure 9 is not clear. DEG can be either upregulated or downregulated compared with a treatment group; the present figure does not depict the pattern of gene expression.

Authors: The Figure identification was rewritten (Figure 1).  (Lines 383)

Reviewer: Line 476-479: Not clear what the authors wanted to state. Diet composition and results of growth performance and carcass traits are not shown in this manuscript.

Authors: We agree with the reviewer. This sentence was removed.

Reviewer 3 Report

Congratulations on preparing this manuscript about "Differential gene expression associated with soybean oil level in the diet of pigs". 

The main strength of this study is using a sufficient number of samples for the study as many RNASeq studies are still using 5 or 6 replicates and the use of 18 samples is a very generous use of resources. The study could be of interest to some readers, but the manuscript could be improved. 

 The manuscript could be improved by using variety of figures as all the figures used currently are of similar nature obtained from single tool. Some examples are, PCA plots from variance stabilized gene count matrix obtained from Deseq2, heatmaps of DEGs etc.

Networks maps (figure 4, 5, 10 ,11) should be improved and should be well explained in the caption as these are taking a lot of space but hard to follow in the current form.

 Some work on proofreading may help as there are several typos, some run-on sentences, and some incomplete sentences. Examples – line 87, line 156, line 175, line 220, line 326, line 479)

The discussion section should be concise. 

Author Response

Piracicaba, São Paulo State, Brazil, June 5, 2022.

Dear Reviewer #3,

      We are returning the revised manuscript entitled “Differential gene expression associated with soybean oil level in the diet of pigs” (Submission ID: 1734763) to Animals for review and possible publication. We sincerely appreciate your comments/suggestions, which were very helpful to improve the quality of our manuscript. Please see below the point-by-point response and the edits in the manuscript were highlighted in yellow. Congratulations on preparing this manuscript about "Differential gene expression associated with soybean oil level in the diet of pigs".

The main strength of this study is using a sufficient number of samples for the study as many RNASeq studies are still using 5 or 6 replicates and the use of 18 samples is a very generous use of resources. The study could be of interest to some readers, but the manuscript could be improved.

Reviewer: The manuscript could be improved by using variety of figures as all the figures used currently are of similar nature obtained from single tool. Some examples are, PCA plots from variance stabilized gene count matrix obtained from Deseq2, heatmaps of DEGs etc.

Authors: The heatmaps were added as Supplemental figures (S1 – C and D).

Reviewer: Networks maps (figure 4, 5, 10,11) should be improved and should be well explained in the caption as these are taking a lot of space but hard to follow in the current form.

Authors: Thank you for your observation. The figures were moved to Supplementary figures (S9, S10, S15, S16) as suggested by reviewer #2. The title was re-written and the DEG was highlighted with red circles.

 Reviewer: Some work on proofreading may help as there are several typos, some run-on sentences, and some incomplete sentences. Examples – line 87, line 156, line 175, line 220, line 326, line 479)

Authors: Thank you for your highlights.

Line 87: The correction was made. (Line 88)

Line 156: The correction was made. (Line 158)

Line 175: The correction was made. (Line 178)

Line 220: The correction was made. (Line 224)

Line 326: The correction was made. (Line 324)

Line 479: The correction was made. (Line 397)

Reviewer: The discussion section should be concise.

Authors: We moved the figures from the main text, which changed the discussion section. Now it is more concise and cleaner.

Round 2

Reviewer 2 Report

Major comment:

1. The authors claim in their manuscript that this is a "nutrigenomic" study. Such claims need to be substantiated with substantial facts, data, and a discussion of the results obtained in this study. I repeat, without the ingredients and the calculated or analysed nutrient composition of the experimental swine diets, it is difficult to evaluate the authors' claim that this is a "nutrigenomic" study. Mere reference to previously published literature does not help the reader make a direct connection between nutrient availability and gene expression results. Furthermore, the authors need to expand the discussion section with nutrient/ diet on gene expression in pigs.

2. Regarding the transcriptome profiling of adipose tissue that was not performed: I do not fully agree with the explanation given by the authors. Budget/finances can never be a scientific argument. I urge the authors to provide more scientific explanations and include them in the manuscript at the appropriate place(s).

Minor comment:

Line 43: Huntington's disease? why this key word? 

Line 406: In vivo to italics

Author Response

Dear Reviewer,

We sincerely appreciate the reviewer’s comments/suggestions, which were very helpful to improve the quality of our manuscript. Please see below the point-by-point response and the edits in the manuscript were highlighted in yellow.

Reviewer #2

Reviewer: The authors claim in their manuscript that this is a "nutrigenomic" study. Such claims need to be substantiated with substantial facts, data, and a discussion of the results obtained in this study. I repeat, without the ingredients and the calculated or analysed nutrient composition of the experimental swine diets, it is difficult to evaluate the authors' claim that this is a "nutrigenomic" study. Mere reference to previously published literature does not help the reader make a direct connection between nutrient availability and gene expression results. Furthermore, the authors need to expand the discussion section with nutrient/ diet on gene expression in pigs.

Authors: Our main concern about it was just presenting the tables from a previously published study. However, we did some modifications and added to this manuscript supplemental tables (S1, S2, S3, and S4). We agree with the reviewer that the readers can follow better our proposal. We added in the discussion section the statement as the reviewer's suggestion. (Lines 970 – 976)

Reviewer: 2. Regarding the transcriptome profiling of adipose tissue that was not performed: I do not fully agree with the explanation given by the authors. Budget/finances can never be a scientific argument. I urge the authors to provide more scientific explanations and include them in the manuscript at the appropriate place(s).

Authors: We agree with the reviewer, then the statement about our decision to focus on skeletal muscle and liver transcriptome study was added to the manuscript. (Lines 1046 – 1052)

Reviewer: Line 43: Huntington's disease? why this key word? 

Authors: We agree with the reviewer's comment. The keyword was removed. (Line: 43)

Reviewer: Line 406: In vivo to italics

Authors: The correction was made. (Line 1213)

Round 3

Reviewer 2 Report

Dear Editor,

The authors have provided reasonable explanation to my concerns mentioned in the previous round.

My overall decision is to Accept after minor revision. Particularly after correcting English language, specifically in Line 401-407. 

Best wishes

Author Response

Dear Reviewer,

We appreciate your advice. The sentence was re-written as suggested and highlighted in yellow. (Lines 507 - 517)

Kind regards,

Prof. Aline Silva Mello Cesar

Department of Agri-Food Industry, Food, and Nutrition – ESALQ/University of São Paulo. Av. Pádua Dias, 11 - Piracicaba - SP, CEP 13418-900; Telephone: +55 (19) 3447-8689